# LASSO-Driven Selection of Biochemical and Clinical Markers for Primary Resistance to PD-1 Inhibitors in Metastatic Melanoma

**DOI:** 10.3390/medicina61091559

**Published:** 2025-08-29

**Authors:** Haydar C. Yuksel, Caner Acar, Gokhan Sahin, Gulcin Celebi, Salih Tunbekici, Burcak S. Karaca

**Affiliations:** 1Department of Medical Oncology, Ege University Medical Faculty, Izmir 35100, Turkey; caner.acar@ege.edu.tr (C.A.); gokhan.sahin@ege.edu.tr (G.S.); burcak.karaca@ege.edu.tr (B.S.K.); 2Division of Internal Medicine, Ege University Medical Faculty, Izmir 35100, Turkey; gulcin.celebi@ege.edu.tr (G.C.); salih.tunbekici@ege.edu.tr (S.T.)

**Keywords:** anti-PD-1 treatment, LASSO, melanoma, primary resistance

## Abstract

*Background and Objectives:* Resistance to immune checkpoint inhibitors (ICIs) reduces treatment efficacy in 40–65% of patients. The ability to predict this at the outset of therapy could help optimise treatment selection and improve patient survival. The aim of this study was to identify factors associated with primary resistance to PD-1 inhibitors in metastatic melanoma and discover predictive markers. *Materials and Methods:* This retrospective study involved 110 patients with non-uveal metastatic melanoma treated with PD-1 inhibitors from 2016 to 2023. Demographic, clinical and haematological data were collected. LASSO regression was utilised to identify the best markers. Bootstrap resampling was performed for internal validation and to overcome overfitting. *Results:* Primary resistance occurred in 44.6% of the patients. The factors associated with resistance included elevated platelet-to-lymphocyte ratio (PLR), the presence of acral/mucosal melanoma, BRAF mutant disease, low globulin levels and ≥3 metastatic sites. An evaluation of the predictive capability of these variables showed robust discrimination, with an area under the receiver operating characteristic curve of 0.831. *Conclusions:* This study identified the key predictors of primary resistance to PD-1 inhibitors to be PLR, globulin levels, metastatic burden and melanoma subtype. These identified parameters may guide the early prediction of primary resistance to PD-1 inhibitors. Future work should externally validate the model and further explore robust strategies to overcome resistance.

## 1. Introduction

Targeted therapies and newly developed immune checkpoint inhibitors (ICIs) have led to significant improvements in the survival of patients with melanoma [1,2,3]. Building on the significant benefits associated with monotherapies, the introduction of combination immunotherapies, such as programmed cell death protein 1 (PD-1) and cytotoxic T-lymphocyte antigen 4 (CTLA-4) inhibitors, has further enhanced the survival outcomes for patients. However, this advancement has been accompanied by a rise in the incidence of side effects [2,4]. Promising data from clinical trials, such as the pivotal Checkmate-067 study, have revealed the clear superiority of combination therapy over PD-1 monotherapy in certain patient subgroups [5]. For instance, patients with BRAF mutant tumours, liver metastases, symptomatic brain metastases and elevated lactate dehydrogenase (LDH) levels showed improved survival outcomes with combination therapies, although the most effective treatment strategies for other subgroups remain a pressing and unresolved issue [6,7].

Patients with certain conditions specific to ICIs, such as hyperprogression and resistance, could derive substantially reduced benefits from these treatments [8,9]. Primary resistance to ICIs during melanoma treatment remains a significant challenge, affecting approximately 40–65% of patients with stage IV melanoma [10]. The concept of primary resistance, which is a significant determinant of the response to ICIs, has been defined as follows by the Immunotherapy Resistance Taskforce that was formed by the Society for Immunotherapy of Cancer: disease progression within six months of ICI therapy after patients have received at least six weeks of ICI monotherapy, with the best response being progressive disease or stable disease [11].

Numerous mechanisms underlying primary resistance have been identified. Tumour-specific factors, such as mutations altering the tumour microenvironment, the tumour mutational burden and programmed death ligand 1 (PD-L1) expression levels, play significant roles in resistance. Additionally, host-related factors, including the status of regulatory T cells (Tregs) and presence of immunosuppressive cytokines (e.g., interleukin 6, interleukin 10, and transforming growth factor beta) secreted by tumour cells, are among the key contributors to resistance [12]. Genetic alterations and receptor-level changes in melanoma have helped us to understand the mechanisms of treatment resistance. Pheomelanin, when exposed to short-wave UV radiation, can create a mutagenic environment. Melanogenesis itself—and particularly its highly reactive intermediates—exhibits cytotoxic, genotoxic, and mutagenic activities, and can induce glycolysis and activation of hypoxia-inducible factor 1-alpha (HIF-1α); this, combined with its immunosuppressive effects, may promote melanoma progression and resistance to immunotherapy [13]. Moreover, non-classical vitamin D receptors, such as RORα and RORγ, have been shown to destabilise cellular homeostasis, further contributing to negative treatment outcomes [14]. Neural crest–derived melanocytes can influence not only the local microenvironment but also systemic homeostasis, thereby modulating the efficacy of immunotherapy. They exert these effects via the hypothalamic–pituitary axis. Through this axis, elevated POMC and catecholamine levels have been observed in patients with metastatic melanoma, and these elevations have been associated with a poor prognosis [15]. Furthermore, in patients with melanoma, BRAF positivity and prior treatment with BRAF-targeted therapies are both associated with a marked reduction in the response to ICIs [16,17,18]. While multi-omics gene analyses and various biomarkers have been utilised to evaluate resistance, these methods remain impractical for routine clinical use [19,20].

Emerging evidence has shown that inflammatory processes are intimately involved in both tumour progression and the response to anticancer treatments [21]. When inflammation becomes chronic, it can induce a profoundly immunosuppressive tumour microenvironment, fuelling the recruitment of myeloid-derived suppressor cells and driving T-cell dysfunction, which can undermine the activity of immune checkpoint inhibitors [22]. Moreover, sustained inflammatory signalling promotes the release of pro-tumourigenic cytokines and angiogenic factors, further hampering effective antitumour immunity. Immunological markers—such as the platelet-to-lymphocyte ratio—and composite scores such as the CALLY index and biomarkers such as fatty acids and protein profiles can serve as useful surrogates for changes in the tumour microenvironment that can be assayed from peripheral blood [23,24,25]. This retrospective study explored the influence of various haematological parameters and clinical features, which have previously been examined for their role in risk stratification, on primary resistance in patients with melanoma. Conducted on a single-centre cohort of patients treated with PD-1 inhibitors, this research evaluated the factors that influence resistance. A novel model based on five factors was developed to predict primary resistance.

## 2. Materials and Methods

### 2.1. Study Population

In this single-centre retrospective study, the biochemical and clinical characteristics of patients who had previously received PD-1 therapy for metastatic melanoma between January 2016 and December 2023 were evaluated to assess treatment resistance. The inclusion criteria for the study were as follows: (i) patients aged over 18 years, (ii) with at least one distant metastasis and (iii) who have undergone at least one assessment according to the Response Evaluation Criteria in Solid Tumors (RECIST) criteria. Patients who received combination ICI therapy, those who died before any assessment could be performed, and patients who had uveal melanoma were excluded from the study.

### 2.2. Data Collection

Demographic, clinical, laboratory and treatment data were extracted from Ege University’s electronic medical records. The baseline clinical characteristics included each patient’s age, gender, Eastern Cooperative Oncology Group (ECOG) performance status, tumour location, tumour staging according to the eighth edition of the TNM classification, treatment regimen and follow-up data. All selected biomarker parameters were obtained from blood test results collected within two weeks prior to treatment initiation. A comprehensive list of the baseline haematological tests is provided in Appendix A. All laboratory parameters were measured in a single institution using standardised protocols, and thus batch effects were not expected. Primary resistance was defined as a tumour response or prolonged stable disease (SD), as per version 1.1 of the RECIST criteria, that lasted six months or longer [11]. The primary outcome measure was tumour response to treatment, which was evaluated using CT imaging at 12 and 16 weeks after the first infusion, and every 12 weeks thereafter. Two treatment outcome groups, responders and primary resistant, were defined using CT imaging following the treatment. Patients who died before radiological evaluations could be performed or who were lost to follow-up were excluded from the analysis. 

### 2.3. Statistical Analysis

Given the lack of prior studies evaluating primary resistance using biochemical markers in this patient cohort, we utilised LASSO regression for variable selection to identify the most predictive parameters. To guard against overfitting, we incorporated bootstrap resampling into our modelling approach. Statistical modelling for the LASSO regression was performed, and the model performance metrics were calculated using R software (version 4.4.2). All other statistical analyses were conducted using Jamovi (Version 2.3.28, The Jamovi Project, 2023, https://www.jamovi.org). Statistical significance was set at *p* ≤ 0.05 for all analyses. Least absolute shrinkage and selection operator (LASSO) regression was performed with the *glmnet* R package (version 4.4.2), utilising an L1 regularisation penalty to shrink the coefficients of each feature to zero and employing 10-fold cross-validation. In addition to clinical characteristics, 23 haematological biomarkers and the related scoring systems were included in the analysis of the study cohort. The calculation methods for all indices included in Table 1 are provided in detail in Appendix B. Internal validation was conducted using the bootstrap method (1000 repetitions).

Logistic regression assumptions, such as linearity between continuous predictors and the logit of the outcome, and absence of multicollinearity among predictors, were tested and adequately met. Linearity was assessed visually and confirmed statistically using the Box-Tidwell test, whereas multicollinearity was evaluated using the variance inflation factor (VIF), with all predictor variables demonstrating VIF values below 2.

### 2.4. Management of Missing Data

Any missing data for the candidate predictor variables in the study cohort below the 5 per cent were addressed using multiple imputation by chained equations using the ***‘survival’***, ***‘survminer’*** and ***‘mice’*** packages. All of the *p*-values were two-sided, with significance defined as *p* < 0.05.

## 3. Results

### 3.1. Patient Characteristics

A total of 110 patients were included in this study. The patients’ characteristics are presented in Table 1. Female patients constituted 46.4% of the cohort, while the median age was 60 years (IQR: 47.2–68.0). Overall, 52.7% of the patients received PD-1 therapy as the first-line therapy. Among those who received ICI treatment in subsequent lines, 23.6% had undergone chemotherapy prior to the immunotherapy, 18.2% had undergone BRAF/MEK inhibitor therapy, and 31.8% had previously received ICIs (mostly CTLA4 inhibitors). Regarding the patients’ clinical characteristics, 38.4% had acral and mucosal melanoma, 56.4% presented with oligometastatic disease and 37.3% were categorised as M1c. In this cohort, primary resistance was absent in 61 patients (55.4%), whereas 49 patients (44.6%) exhibited primary resistance. There were no statistically significant differences between these two groups of patients regarding their baseline characteristics, including age, sex, ECOG performance status, prior BRAF inhibitor therapy, chemotherapy or previous treatment with other ICIs (*p* > 0.05 for all).

### 3.2. Markers Selection

All of the available clinical indicators, including the clinicopathological features and biomarkers (Table 1), were subjected to LASSO Cox regression to check for significant correlation with primary resistance. The optimal lambda value was selected using 10-fold cross-validation. The variables identified through the LASSO regression were then incorporated into the multivariate logistic regression concerning primary resistance. For the logistic regression models, multicollinearity was assessed using the variance inflation factor (VIF) values, all of which were confirmed to be below 10. Twelve parameters were demonstrated to contribute to the model’s prediction accuracy (Appendix C). A backward stepwise logistic regression analysis was performed to find the best model, which revealed that lesion localisation, BRAF mutation status, metastatic burden, PLR, and globulin levels were significant predictors of primary resistance. To further clarify the predictor contributions and interdependencies, we generated a range-adjusted feature importance bar plot (Figure A2), which shows us PLR and serum globulin emerged as the two strongest contributors to our logistic regression model.

### 3.3. Construction of Nomogram

Logistic regression was applied to finalise the parameters associated with primary resistance based on those identified through LASSO analysis. Upon evaluation, globulin levels, having more than three metastases, BRAF mutation, and acral melanoma were found to be associated with resistance. (Table 2) The parameters identified from this analysis were visualised in a nomogram (Figure 1). Multicollinearity among the five predictors was evaluated in two ways. First, we generated a predictor correlation heat-map (Figure A1), which showed that all pairwise Pearson correlation coefficients were below |r| = 0.7. Second, we calculated variance inflation factors (VIFs) for the logistic regression model; all VIF values fell below 2.0, confirming that multicollinearity was not a concern in our analysis. A visual assessment of potential outliers was performed using boxplots for all continuous variables. No extreme outliers were detected beyond the interquartile range thresholds, and all observed values were within acceptable clinical and statistical limits. Therefore, no data points were excluded on the basis of outlier status.

### 3.4. Model Performance Metrics

A total of 110 observations were included in the analysis after imputing the missing data. The discriminative ability of the model was assessed using the C-statistic (area under the receiver operating characteristic [ROC] curve) (Figure 2), which was found to be 0.831, indicating strong classification performance. The nomogram model demonstrated a higher net benefit across a range of thresholds, suggesting its clinical utility in guiding treatment decisions (Figure 3A). The calibration curve shows the agreement between the predicted probabilities and observed outcomes. The bias-corrected curve (solid line) closely aligned with the ideal calibration line (dashed line), indicating a good predictive accuracy for the nomogram model. The mean absolute error was 0.026, confirming the model’s reliability (Figure 3B). The goodness-of-fit of the logistic regression model was assessed using the Hosmer–Lemeshow test. The test indicated no evidence of a lack of fit (χ^2^ = 5.03, df = 8, *p* = 0.754), thereby demonstrating good agreement between the predicted and observed values. Additionally, the Brier score of the model was calculated to be 0.165, further supporting its validity.

## 4. Discussion

In this study, we evaluated a wide array of variables with the potential to predict primary resistance, with a particular emphasis on the contributions of biochemical parameters. The statistical analysis demonstrated that lesion localisation, BRAF mutation status, metastatic burden, platelet-to-lymphocyte ratio (PLR) and serum globulin levels were all significant predictors of resistance. Moreover, by applying rigorous statistical methods, we identified the markers exhibiting the highest predictive accuracy.

The individualisation of cancer treatment processes is crucial for optimising therapeutic responses in contemporary clinical practice. The variability in disease behaviours and patient responses to treatment necessitates employing diverse methods to select the most effective regimen at the start of treatment. In this study, we aimed to identify the patient subgroup exhibiting primary resistance to PD-1 inhibitors in metastatic melanoma. By combining patient characteristics and biochemical parameters, we aimed to investigate clinical and hematologic parameters associated with primary resistance to PD-1 inhibitors and to assemble them into a nomogram framework for hypothesis generation and future validation, rather than making definitive claims about immediate clinical implementation

Since the development of therapies targeting anti-PD-1 inhibitors, numerous mechanisms underlying resistance and a variety of related biomarkers have been proposed. From the initiation of anti-PD-1 treatment, factors such as tumour mutational burden, specific mutations (e.g., STK11, KEAP), clonal architecture, aneuploidy, immune-escape mechanisms, and interferon-γ signalling pathways have all been linked to resistance [26]. However, elucidating these mechanisms is often challenging. Artificial intelligence–based multi-omics approaches have recently gained momentum [27], and integrating multiple resistance-associated genes identified by multi-omics with AI-driven machine-learning models has demonstrated high predictive accuracy for resistance [28,29].

Alongside these complex approaches, simpler clinical predictors for melanoma resistance have also been proposed. In a multicentre study by van Duin et al., LDH level, patient sex, ECOG performance status, systemic therapy type and melanoma site, presence of satellite lesions at diagnosis, and time to metastasis were all associated with resistance [30]. Nosrati et al. identified LDH, age, sex, and liver metastasis as key predictors for primary resistance [31]. Pires da Silva’s model for objective response rate included WHO performance status, LDH, presence of liver and lung metastases, treatment type, and line of therapy [7]. Although these models achieved high AUCs in their own validation cohorts, their performance in external populations remains unclear. Furthermore, a high rate of missing data prevented evaluation of some parameters, such as serum globulin, in these analyses.

Numerous molecular and systemic mechanisms underlie the development of primary resistance. These include tumour heterogeneity driven by pre-existing mutations that permit the clonal expansion of resistant cells, therapy-induced genetic alterations accompanying disease progression, the emergence of a mutator phenotype, metabolic diversity within the tumour, and dynamic shifts in both the tumour microenvironment and systemic homeostasis [13,14,15]. Broadly speaking, factors contributing to resistance fall into three categories: (i) failure to elicit robust anti-tumour T-cell responses, (ii) suboptimal activity of tumour-specific T cells, and (iii) defective generation of effector memory T cells [32]. Identifying which patients manifest these traits can be challenging. Although histopathological and genomic analyses are invaluable for detecting resistance mechanisms, their application in routine clinical practice is often limited by logistical and practical constraints. Studies have demonstrated that the presence of a BRAF mutation and the use of BRAF inhibitors are both associated with significantly worse treatment responses to ICIs in patients with metastatic melanoma [5,17]. For this reason, various guidelines recommend combination therapies for this patient group to achieve better treatment outcomes [33]. The pivotal DREAMseq trial showed that the combination of nivolumab/ipilimumab, followed by BRAF and MEK inhibitor therapy, was associated with a better treatment response [34]. The alteration of the immune microenvironment by anti-BRAF therapy is suggested to be a potential cause of this phenomenon [35,36]. In studies evaluating resistance in patients with BRAF mutations, the primary resistance rates were observed to be 36% in mutant-BRAF tumours and 33% in wild-type-BRAF tumours when PD-1 inhibitors were used as the first-line therapy. In later treatment lines, these rates were found to be 56.4% and 51.3%, respectively [37,38,39,40]. Among patients receiving BRAF/MEK therapy, the primary resistance rate was found to be 56.41% [40]. Also, it has been demonstrated that BRAF-mutant patients exhibit a higher frequency of resistance-related genes, such as PTEN. This observation might serve as an indirect indicator for the presence of other resistance-associated genes [35].

The relationship between globulin levels and resistance has not previously been evaluated. However, a correlation has been observed between the albumin-to-globulin ratio (AGR) and treatment response, where a low AGR was associated with a reduced response to ICIs [41,42,43,44]. This phenomenon has been primarily attributed to the impaired metabolism caused by liver dysfunction and anti-TNF13B responses [45]. Despite this, we believe that equating the concept of resistance, as assessed in this study, with treatment response may not be entirely appropriate. Notably, a positive correlation has been demonstrated between the globulin level and the expression of PD-1, an indicator of resistance. This may be related to resistance through the relationship between PD-1 and PD-L1 [46], although further research is needed to explore this connection. Moreover, as this finding contradicts the findings reported in the literature, it is necessary to evaluate this finding in relation to resistance in future studies. Additionally, immunoglobulin (Ig) components of globulin have been shown to influence survival. Elevated levels of IgG2 and IgG4 have been associated with improved responses to immune checkpoint inhibitors (ICIs). This association might be due to antibodies developed against melanoma-associated antigens such as TRP1, TRP2, and gp100 [47].

The localisation of metastases is a critical parameter influencing the response to ICIs in patients with melanoma. Studies have demonstrated that the presence of brain, bone and liver metastases is associated with lower objective response rates and shorter progression-free survival [48,49]. Additionally, the presence of metastases in more than two sites is linked to treatment resistance [50,51,52]. However, the evidence supporting these findings remains limited, with few robust studies available in the literature. Notably, liver metastases have been observed to impair antigen presentation and foster the development of an immunosuppressive tumour microenvironment, further complicating therapeutic responses [53]. In this study, the number of metastatic sites was identified as a more critical predictor of the treatment response compared to the presence of liver or other site-specific metastases when analysed in the context of the disease stage. Through leveraging the strengths of the LASSO analysis, particularly its ability to address collinearity and prioritise significant variables, the number of metastatic sites emerged as a more clinically relevant parameter. This underscores the importance of evaluating the extent of the metastatic burden rather than focusing solely on specific metastatic locations in clinical decision-making.

The distinct characteristics of melanomas in different anatomical locations contribute to the variations in treatment responses. Acral and mucosal melanomas are associated with a low tumour mutational burden, which is considered a factor that drives resistance [52]. Additionally, CCND1 amplification and CDKN2A loss—which are common genetic alterations seen in acral melanoma—can activate the CDK4 pathway, promoting primary resistance to therapy [54]. In this study, treatment resistance was observed more frequently in patients diagnosed with acral and mucosal melanoma.

High PLRs were associated with increased platelet counts and decreased lymphocyte counts. Previous studies have demonstrated that elevated PLRs are correlated with worse survival outcomes and treatment resistance [55]. A similar association has also been observed in melanoma patients [56]. When evaluating the connection between high platelet levels and resistance, it has been shown that platelets can shield tumour cells from immune responses mediated by NK cells. Moreover, recent research has indicated that genetic manipulation of platelets or the use of antiplatelet pharmacological agents may enhance the effectiveness of adaptive T-cell therapies for cancer by targeting the GARP-TGFβ axis [57,58]. Moreover, studies have linked low lymphocyte counts to poor treatment responses [59]. The worse outcomes observed in patients with lymphopenia could be attributed to alterations in the tumour microenvironment. The accumulation of myeloid-derived suppressor cells, type 2 macrophages or regulatory T cells, along with the production of immunosuppressive cytokines and metabolites, may contribute to tumour progression [60]. In this study, the observation that higher PLRs correlate with poor responses aligns with the literature and contributes to the model.

What steps can we take to overcome resistance? First and foremost, combination therapies play a critical role in reducing primary resistance rates. While the primary resistance rates for anti-PD-1 therapies range between 40% and 60%, they have been observed to decrease to 30–40% with anti-PD-1 and CTLA-4 combination therapies [10,61]. Utilising ICI combination therapies, tumour-infiltrating lymphocytes (TILs), and mRNA vaccines could be a promising strategy for optimising treatment outcomes in patients identified as resistant. At the same time, the use of monotherapy in patients without resistance may help to ensure efficacy while avoiding side effects.

## 5. Limitation

The principal limitation of this study is the lack of external validation owing to its single-centre, retrospective design. Also, because all data were generated under homogeneous conditions, no outlier removal or batch-effect adjustment was applied, which may limit generalizability. Additionally, the relatively small patient cohort resulted in a heterogeneous population and uneven subgroup distributions, which may have led to reduced statistical significance. To address these issues, future research should re-evaluate the current nomogram in larger, multicentre populations.

## 6. Conclusions

In this study, parameters associated with primary resistance were evaluated in patients with metastatic melanoma who were receiving PD-1 therapy. These parameters were selected by using LASSO and logistic regression analyses to develop the most accurate model possible, which was then transformed into a nomogram. Novel markers, not previously reported in the literature, have been identified in this study. Based on our findings, this nomogram is less a decision-making algorithm and more a tool to help clinicians predict the benefits patients are likely to derive from their current treatment. The clinical applicability of these markers will become clearer following their validation in larger patient cohorts.

## Figures and Tables

**Figure 1 medicina-61-01559-f001:**
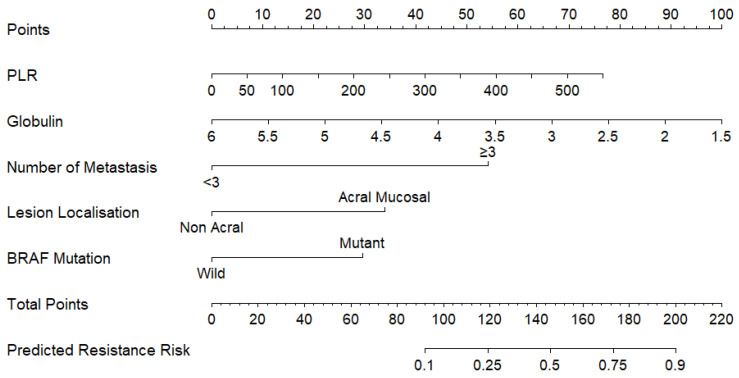
Nomogram for predicting primary resistance to anti–PD-1 therapy in metastatic melanoma. For each predictor, draw a vertical line to the Points scale and record the score. Sum these scores to obtain the Total Points, then project down to the Predicted Resistance Risk axis to read the estimated probability. Higher Total Points correspond to a greater risk of resistance. For the calculation, more than three metastatic sites add 53 points, acral/mucosal localisation adds 34 points, and BRAF-mutant status adds 29 points; globulin and PLR should be scored according to their levels.

**Figure 2 medicina-61-01559-f002:**
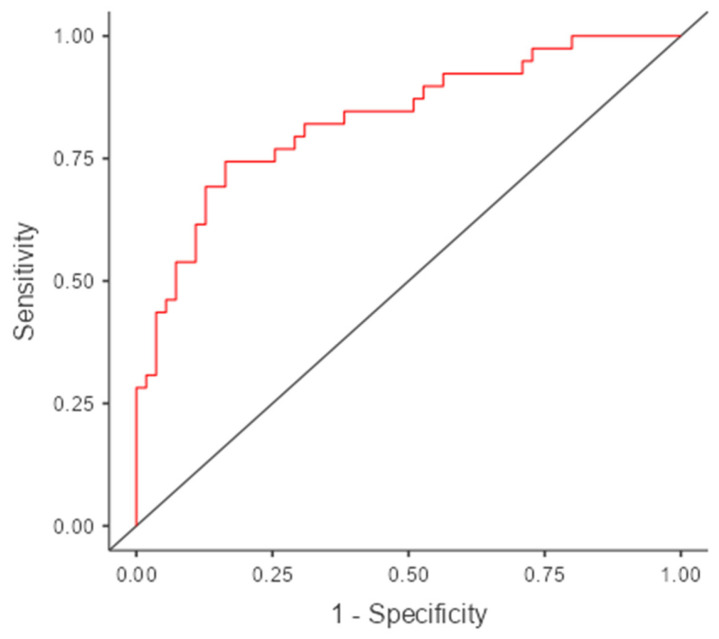
ROC curve for the nomogram model (AUC = 0.831).

**Figure 3 medicina-61-01559-f003:**
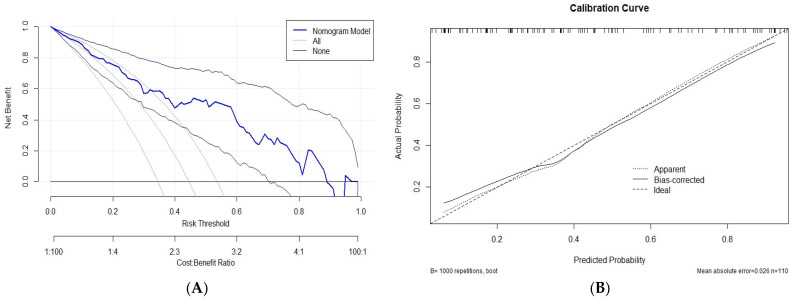
(**A**) Decision curve analysis for the nomogram model predicting resistance to PD-1 inhibitors. (**B**) Calibration curve for the nomogram model predicting resistance to PD-1 inhibitors.

**Table 1 medicina-61-01559-t001:** Clinical and demographic characteristics and haematological parameters in the study cohort.

		Resistance Status		
Patient Characteristics		Non-Resistant	Resistant	Total
Total N (%)		61 (55.5)	49 (44.5)	110
Age	Median (IQR)	61.0 (48.0–68.0)	59.0 (46.0–67.0)	60.0 (47.2–68.0)
Sex	Male	34 (55.7)	24 (49.0)	58 (52.7)
	Female	27 (44.3)	25 (51.0)	52 (47.3)
ECOG PS	Poor (0–1)	5 (8.2)	8 (16.3)	13 (11.8)
	Good (2–3)	56 (91.8)	41 (83.7)	97 (88.2)
BRAF/MEKi treatment before PD1	No	51 (83.6)	39 (79.6)	90 (81.8)
	Yes	10 (16.4)	10 (20.4)	20 (18.2)
Chemotherapy Before PD1	No	48 (78.7)	36 (73.5)	84 (76.4)
	Yes	13 (21.3)	13 (26.5)	26 (23.6)
ICI before ICI	No	40 (65.6)	35 (71.4)	75 (68.2)
	İpilimumab	18 (29.5)	14 (28.6)	32 (29.1)
	PD1	3 (4.9)	0 (0.0)	3 (2.7)
Metastasis Status	De Novo	28 (45.9)	15 (30.6)	43 (39.1)
	Recurrent	33 (54.1)	34 (69.4)	67 (60.9)
Anti-PD1 Treatment Line	First	35 (57.4)	23 (46.9)	58 (52.7)
	Second and Beyond	26 (42.6)	26 (53.1)	52 (47.3)
Lesion Localization	Non-Acral Cutaneous	44 (72.1)	24 (49.0)	68 (61.8)
	Acral	9 (14.7)	15 (30.5)	24 (21.8)
	Mucosal	8 (13.1)	10 (20.4)	18 (16.4)
Oligometastasis Status	No	26 (42.6)	36 (73.5)	62 (56.4)
	Yes	35 (57.4)	13 (26.5)	48 (43.6)
Stage	M1a	26 (42.6)	8 (16.3)	34 (30.9)
	M1b	14 (23.0)	10 (20.4)	24 (21.8)
	M1c	17 (27.9)	24 (49.0)	41 (37.3)
	M1d	4 (6.6)	7 (14.3)	11 (10.0)
Metastasis Site	<3	49 (80.3)	20 (40.8)	69 (62.7)
	≥3	12 (19.7)	29 (59.2)	41 (37.3)
BRAF mutation	Wild	47 (77.0)	34 (69.4)	81 (73.6)
	Mutant	14 (23.0)	15 (30.6)	29 (26.4)
SIPS	Good	50 (86.2)	33 (80.5)	83 (83.8)
	Poor	8 (13.8)	8 (19.5)	16 (16.2)
Royal Marsden Hospital Score	Good	29 (50.9)	10 (24.4)	39 (39.8)
	Poor	28 (49.1)	31 (75.6)	59 (60.2)
MD Anderson ICI Score	Good	17 (28.8)	5 (11.9)	22 (21.8)
	Intermediate	38 (64.4)	26 (61.9)	64 (63.4)
	Poor	4 (6.8)	11 (26.2)	15 (14.9)
Lymphocyte	Median (IQR)	1820.0 (1315.0–2315.0)	1430.0 (1080.0–1780.0)	1635.0 (1255.0–2125.0)
Monocyte	Median (IQR)	560.0 (440.0–785.0)	560.0 (439.5–710.0)	560.0 (440.0–737.5)
Eosinophil	Median (IQR)	180.0 (102.5–290.0)	100.0 (50.0–205.0)	130.0 (60.0–250.0)
Albumin	Median (IQR)	4.3 (4.0–4.6)	4.1 (3.9–4.4)	4.2 (3.9–4.5)
Globulin	Median (IQR)	2.8 (2.5–3.2)	2.8 (2.4–3.0)	2.8 (2.5–3.1)
CRP	Median (IQR)	0.2 (0.1–1.1)	0.5 (0.2–1.4)	0.3 (0.1–1.4)
LDH	Median (IQR)	201.0 (175.0–274.0)	227.0 (177.0 t–290.0)	208.5 (175.0–278.2)
MPV	Median (IQR)	10.1 (9.6–10.7)	9.8 (9.2–10.3)	10.0 (9.5–10.7)
NLR	Median (IQR)	2.4 (1.7–3.3)	3.0 (2.5–3.7)	2.7 (2.0-3.6)
LMR	Median (IQR)	3.0 (2.5–4.2)	2.6 (2.0–3.3)	2.9 (2.2- 3.8)
PLR	Median (IQR)	153.3 (112.2–196.9)	196.4 (127.0–294.2)	162.9 (113.8–236.8)
MPV/Lymphocyte	Median (IQR)	5.4 (4.6–7.5)	6.5 (5.7–8.8)	6.0 (4.7–8.3)
HALP Score	Median (IQR)	39.1 (26.8–50.3)	26.4 (16.0–49.2)	34.3 (19.9–50.2)
PIV Score	Median (IQR)	383.3 (206.1–651.0)	474.0 (276.6–936.6)	438.1 (225.7–822.8)
PNI Score	Median (IQR)	51.6 (46.5–54.7)	48.7 (44.5–51.6)	50.2 (45.6–54.4)
SII Score	Median (IQR)	676.0 (421.2–982.4)	889.7 (526.4–1365.1)	724.2 (465.3–1123.7)

Data are reported as median (interquartile range) for continuous variables and frequencies (percentages) for categorical variables. Abbreviation: ECOG-PS: Eastern Cooperative Oncology Group (ECOG) performance status, ICI: immune checkpoint inhibitors, SIPS: Scottish Inflammatory Prognostic Score, CRP: C Reactive Protein, LDH: Lactate Dehydrogenase, NLR: Neutrophil-to-Lymphocyte Ratio, LMR: Lymphocyte-to-Monocyte Ratio, PLR: Platelet-to-Lymphocyte Ratio, MPV: Mean Platelet Volume, LDH: Lactat Dehidrogenaz, CII: Combine Immun Index, HALP Score: Haemoglobin, Albumin, Lymphocyte, Platelet Score, PIV: Pan-Immune-Inflammation Value, PNI: Prognostic Nutritional Index, SII: Systemic Immune-Inflammation Index.

**Table 2 medicina-61-01559-t002:** Univariate and multivariate regression analysis of the indicators of primary resistance.

Primary Resistance		No	Yes	OR (Univariable)	OR (Multivariable)
MPV/lymphocyte	Mean (SD)	6.4 (2.8)	7.8 (3.8)	1.16 (1.01–1.35, *p* = 0.043)	
PNI	Mean (SD)	51.9 (5.8)	48.7 (5.8)	0.91 (0.84–0.98, *p* = 0.015)	
ICI before ICI	No	40 (53.3)	35 (46.7)	-	
	Yes	21 (60)	14 (40)	0.59 (0.22–1.47, *p* = 0.264)	
Metastasis Status	Denovo	28 (65.1)	15 (34.9)	-	
	Recurrent	33 (49.3)	34 (50.7)	2.28 (0.97–5.63, *p* = 0.065)	
Oligometastasis Status	No	49 (71.0)	20 (29.0)	-	
	Yes	12 (29.3)	29 (70.7)	0.30 (0.12–0.72, *p* = 0.008)	
Lesion Localisation	Non Acral	44 (64.7)	24 (35.3)	-	
	Acral + Mucosal	17 (40.5)	25 (59.5)	2.32 (0.99–5.53, *p* = 0.055)	3.82 (1.16–12.6, *p* = 0.027)
BRAF Mutation	Wild	28 (65.1)	15 (34.9)	-	-
	Mutant	33 (49.3)	34 (50.7)	1.78 (0.69–4.65, *p* = 0.234)	5.22 (1.31–20.82, *p* = 0.019)
Metastasis Grup	<3	47 (58.0)	34 (42.0)	-	
	≥3	14 (48.3)	15 (51.7)	5.75 (2.35–14.90, *p* < 0.001)	11.4 (3.43–37.55, *p* < 0.001)
PLR	Mean (SD)	164.2 (72.5)	223.7 (118.3)	1.01 (1.00–1.01, *p* = 0.006)	1.007 (1.01–1.01, *p* = 0.019)
Globulin	Mean (SD)	2.9 (0.7)	2.8 (0.5)	0.75 (0.36–1.49, *p* = 0.429)	0.35 (0.14–0.88, *p* = 0.027)

Abbreviation: MPV: Mean Platelet Volume, ICI: immune checkpoint inhibitors PLR: Platelet-to-Lymphocyte Ratio.

## Data Availability

The data that support the findings of this study are available from the corresponding author upon reasonable request.

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
