# Peer review of "LASSO-Driven Selection of Biochemical and Clinical Markers for Primary Resistance to PD-1 Inhibitors in Metastatic Melanoma"

_medicina, 2025, doi:10.3390/medicina61091559_

Round 1
Reviewer 1 Report
Comments and Suggestions for Authors
The manuscript aims to develop a nomogram to predict primary resistance to PD-1 inhibitors in patients with metastatic melanoma, utilizing clinical and laboratory features. While this is an important and timely topic, several significant limitations significantly reduce the impact, reliability, and generalizability of the findings.
Major Concerns:
1 The dataset appears to be small and lacks both internal cross-validation and external validation. Predictive modeling without validation cannot be considered robust or clinically useful. The absence of a validation cohort increases the risk of overfitting and raises questions about the reliability of the proposed nomogram.
Data Selection and Variable Justification
The rationale behind selecting variables included in the nomogram is unclear. Were they based on clinical relevance, statistical significance, or prior literature? There is no mention of how multicollinearity or redundancy between predictors was handled.
2 The manuscript does not provide sufficient methodological detail. It is unclear how missing data were handled, whether assumptions of logistic regression were tested, and how performance metrics (AUC, calibration, etc.) were assessed. The lack of clarity on these core issues weakens the manuscript’s credibility.
Overinterpretation of Results
The conclusions make strong claims about clinical utility without adequate support. Without prospective data or at least external validation, such claims are premature. Statements suggesting clinical applicability should be reworded as exploratory.
3 The single figure lacks adequate annotation. Axis labels are small, and critical terms such as “risk score” or “points” are not clearly defined. A proper calibration plot or decision curve analysis would enhance the utility of the nomogram.
4 The introduction and discussion miss several key references on immune resistance mechanisms and predictive modeling in immunotherapy. These should be incorporated to position the work within the current scientific landscape.
The concept of predicting PD-1 resistance is valuable, but the current study lacks the statistical power, methodological rigor, and validation required to support its claims.
Comments on the Quality of English LanguageThe English throughout the manuscript needs careful editing for grammar, clarity, and professional tone. Several sections are redundant or vaguely phrased, making interpretation difficult.
Author Response
Dear Reviewer,
We sincerely appreciate your interest in our study and your invaluable comments. Your meticulous and constructive feedback has allowed us to enhance the quality of our manuscript at every stage, from methodological design to the interpretation of results. In particular, the additions clarifying the underlying biological mechanisms and the emphasis on findings that contribute to the scientific literature have strengthened our work and underscored its significance. In response, we have carefully implemented all of your suggestions, enriching both the Introduction and Discussion sections. Your insightful comments not only deepened our explanation of the biological rationale but also highlighted the nomogram’s predictive accuracy and clinical utility, thereby reinforcing its potential impact on patient stratification in metastatic melanoma.
Comment 1.The dataset appears to be small and lacks both internal cross-validation and external validation. Predictive modeling without validation cannot be considered robust or clinically useful. The absence of a validation cohort increases the risk of overfitting and raises questions about the reliability of the proposed nomogram.
Response 1. External validation often enhances a nomogram’s credibility, but our single-center cohort of a relatively rare malignancy like melanoma was too small to permit such an approach. To guard against overfitting, we implemented bootstrap resampling for internal validation and employed a penalized Cox regression model using LASSO. Nonetheless, future external validation in multicenter studies will be essential to confirm and generalize the nomogram’s predictive performance.
Comment 2.The rationale behind selecting variables included in the nomogram is unclear. Were they based on clinical relevance, statistical significance, or prior literature? There is no mention of how multicollinearity or redundancy between predictors was handled.
Response 2.To our knowledge, no prior studies have investigated primary resistance using routine biochemical parameters. While early‐mortality prognostic tools such as the Royal Marsden Hospital (RMH) and MD Anderson scores have been applied in this context, they do not fully capture the biological mechanisms underlying resistance. As a result, direct comparisons to these established indices could not be included in the Introduction or Discussion. Instead, the Discussion emphasizes relevant clinic-pathological features supported by the literature. Given the exploratory nature of our work, variables selected via LASSO and confirmed as independent predictors in multivariate Cox regression were incorporated into the nomogram. A variance inflation factor threshold of <5 was imposed to guard against collinearity, and the finalized risk‐score table was presented as a nomogram within the manuscript.
Comment 3.The manuscript does not provide sufficient methodological detail. It is unclear how missing data were handled, whether assumptions of logistic regression were tested, and how performance metrics (AUC, calibration, etc.) were assessed. The lack of clarity on these core issues weakens the manuscript’s credibility.
Response 3: Missing values affecting fewer than 5% of the cohort were imputed using multiple imputation within the Cox proportional hazards framework. Nomogram-specific performance metrics have also been incorporated into the revised manuscript.
Comment 4.Overinterpretation of Results: The conclusions make strong claims about clinical utility without adequate support. Without prospective data or at least external validation, such claims are premature. Statements suggesting clinical applicability should be reworded as exploratory.
Response 4.The usage recommendations and accompanying comments have been revised in accordance with the reviewer’s suggestions.
Comment 5. The single figure lacks adequate annotation. Axis labels are small, and critical terms such as “risk score” or “points” are not clearly defined. A proper calibration plot or decision curve analysis would enhance the utility of the nomogram.
Response 5. Based on the reviewer’s feedback, we have updated the nomogram’s performance metrics and added both decision‐curve and calibration‐curve analyses to the manuscript.
Comment 6.The introduction and discussion miss several key references on immune resistance mechanisms and predictive modeling in immunotherapy. These should be incorporated to position the work within the current scientific landscape.
Response 6. In response to the reviewer’s recommendations, both the Introduction and Discussion have been expanded. The Introduction now includes a new paragraph outlining immune-resistance mechanisms, and additional commentary on the mechanistic basis of our findings has been added.

Reviewer 2 Report
Comments and Suggestions for Authors
In the following work, Yuksel et al. study factors associated with primary resistance to PD-1 inhibitors in metastatic melanoma and discover predictive markers. To this end, evaluate hematological parameters and clinical characteristics of patients, using various statistical methods.
Minor points
- On line 4 of the Introduction section, there's a number 4 in the middle of the word "programmed." It should be removed.
- Authors should propose short- and medium-term perspectives that will overcome the limitations of the work. These limitations are described by the authors at the end of their manuscript.
Author Response
Dear Reviewer,
We sincerely appreciate your interest in our study and your invaluable comments. Your meticulous and constructive feedback has allowed us to enhance the quality of our manuscript at every stage, from methodological design to the interpretation of results. In response, we have carefully implemented all of your suggestions, enriching limitation and future perspective. Your insightful comments not only deepened our explanation of the biological rationale but also highlighted the nomogram’s predictive accuracy and clinical utility, thereby reinforcing its potential impact on patient stratification in metastatic melanoma.
Comments 1: Authors should propose short- and medium-term perspectives that will overcome the limitations of the work. These limitations are described by the authors at the end of their manuscript.
Response 1: The absence of external validation was identified as the principal limitation of the current study; accordingly, we plan to conduct multicenter investigations with larger patient cohorts to address this issue.

Reviewer 3 Report
Comments and Suggestions for Authors
The manuscript presented for review is focused on retrospective study of selection of the key markers that could serve as predictors of primary resistance of patients with non-uveal metastatic melanoma to immunotherapy with immune checkpoint inhibitors (ICIs). The work is relevant and has been performed using adequate statistical methods.
The “Introduction” section presents literature data on therapeutic efficacy of the treatment of patients with metastatic melanoma using ICIs, also on primary resistance to this therapy. It should be noted that the overwhelming majority of cited works are dated to the last five years.
This makes the relevance of the presented study clearly justified.
The section "Materials and Methods", which is provided with Appendix 1, contains all necessary information about the set of assessed indices and physiological parameters. The used statistical methods are described clearly and can be reproduced by other researchers.
The obtained results are described in detail, and the section includes relevant Tables that facilitate data analysis and understanding of the research.
The "Discussion" section is of special interest. It contains a detailed discussion of all indices that the authors identify as possible predictors of primary resistance to immunotherapy compared to the data from modern literature. Suggestions how to overcome primary resistance to ICIs therapy are also included into this section.
Certainly the work has a drawback which the authors are fully aware of, i.e., the sample of only 110 patients who were examined and treated in one medical center. However the used statistical data processing methods have allowed the authors to obtain reliable results for this group of patients. Still it is desirable in future to perform a multicenter study and significantly expand the group of examined patients. The study is of interest to oncologists, immunologists, and hematologists.
The manuscript may be improved after minor correction of punctuation mistakes.
Author Response
Comment 1: Certainly the work has a drawback which the authors are fully aware of, i.e., the sample of only 110 patients who were examined and treated in one medical center. However the used statistical data processing methods have allowed the authors to obtain reliable results for this group of patients. Still it is desirable in future to perform a multicenter study and significantly expand the group of examined patients. The study is of interest to oncologists, immunologists, and hematologists.
Responce 1: Dear Reviewer,
Thank you for your thorough and insightful evaluation of our manuscript. We are grateful for your recognition of the study’s relevance and the rigor of our statistical methods. In response to your suggestions, we have carefully corrected all minor punctuation errors throughout the text and ensured consistency in formatting. We have also reaffirmed the need for future multicenter investigations to expand our single-center cohort, as well as clarified this limitation in the Discussion. Your constructive feedback has strengthened both the clarity and reproducibility of our work, and we sincerely appreciate the time and expertise you have devoted to improving our manuscript.

Round 2
Reviewer 1 Report
Comments and Suggestions for Authors
The revised manuscript proposes a predictive nomogram for primary resistance to PD-1 inhibitors in metastatic melanoma using retrospective clinical and hematologic parameters. While the study addresses a timely and clinically relevant question, several critical issues raised in the prior review remain only partially resolved:
1. The model was developed and tested within a single-institution retrospective dataset without external validation. Although bootstrap internal validation was added, it does not sufficiently address concerns about overfitting or generalizability. Without an independent validation cohort, claims of clinical applicability remain premature and should be clearly tempered throughout the manuscript.
2. The statistical process for variable selection has been improved through LASSO regression and multicollinearity checks; however, the biological and clinical rationale for the final predictors (e.g., globulin, BRAF status) remains underdeveloped. The manuscript would benefit from a clearer explanation of why these variables are mechanistically or prognostically meaningful in the context of immunotherapy resistance.
3. Several important details are still missing or insufficiently described. Specifically:
-Handling of outliers and batch effects is not mentioned.
-The timing of biomarker collection relative to treatment initiation is unclear.
-The definition of primary resistance (based on RECIST) would benefit from additional clarity, including radiologic review processes and censoring rules.
-Assumptions underlying logistic regression (e.g., linearity of the logit, absence of multicollinearity) should be reported explicitly.
4. Figures, particularly the nomogram and calibration plots, remain difficult to interpret. Labels such as “points” and “risk score” are undefined, and the legends lack sufficient descriptive detail. Incorporating a summary visualization of variable contributions (e.g., a feature importance plot or heatmap) would improve clarity and accessibility.
5. Several statements in the Abstract, Results, and Conclusion imply the model’s clinical readiness. Given the exploratory nature of the analysis and the absence of prospective or external validation, these claims should be reworded more cautiously. The model may offer insights into hypothesis generation but should not be presented as a decision-support tool at this stage.
6. The revised Discussion includes additional references but still omits relevant recent studies on machine learning models and immune biomarker prediction in melanoma. Including these would better position the work within the current scientific landscape and acknowledge ongoing efforts in this domain.
Comments on the Quality of English Language
The quality of English in the manuscript is generally acceptable but requires moderate revision to improve clarity and precision. Several sections contain grammatical errors, awkward sentence structures, and repetitive phrasing that detract from the overall readability. In particular, descriptions of statistical methods and clinical interpretations would benefit from clearer wording and more consistent use of terminology. Additionally, transitions between sections can be improved to enhance logical flow. A thorough language edit by a native English speaker or professional editor is recommended to ensure clarity, coherence, and appropriate scientific tone throughout the manuscript.
Author Response
Dear Reviewer,
Thank you very much for taking the time to re‐evaluate our manuscript and for your invaluable, constructive comments. Your careful review and insightful suggestions have greatly enhanced the clarity and rigor of our work. We are also grateful for your support of our study’s publication, and we appreciate the opportunity to address your concerns and strengthen the manuscript accordingly.
- The model was developed and tested within a single-institution retrospective dataset without external validation. Although bootstrap internal validation was added, it does not sufficiently address concerns about overfitting or generalizability. Without an independent validation cohort, claims of clinical applicability remain premature and should be clearly tempered throughout the manuscript.
Given the limited size of our current dataset, we share the reviewers’ concerns regarding generalizability. Accordingly, all statements in the manuscript pertaining to the model’s clinical applicability have been tempered and are now presented as hypothesis-generating
- The statistical process for variable selection has been improved through LASSO regression and multicollinearity checks; however, the biological and clinical rationale for the final predictors (e.g., globulin, BRAF status) remains underdeveloped. The manuscript would benefit from a clearer explanation of why these variables are mechanistically or prognostically meaningful in the context of immunotherapy resistance.
In the Discussion, we have expanded the contextualization of each parameter included in the nomogram and incorporated additional, recent studies on immune–hematologic predictors of PD-1 resistance. However, the specific role of serum globulin in mediating both primary resistance and overall survival remains poorly characterized in the existing literature. Consequently, we found it challenging to draw definitive mechanistic links between elevated globulin levels and treatment failure. To address this gap, we have explicitly noted in the revised Discussion that “given the paucity of data on globulin’s prognostic significance in the context of PD-1 inhibitor resistance and patient survival, further prospective and translational studies are required to clarify its biological and clinical relevance.
- Several important details are still missing or insufficiently described. Specifically:
-Handling of outliers and batch effects is not mentioned.
-The timing of biomarker collection relative to treatment initiation is unclear.
-The definition of primary resistance (based on RECIST) would benefit from additional clarity, including radiologic review processes and censoring rules.
-Assumptions underlying logistic regression (e.g., linearity of the logit, absence of multicollinearity) should be reported explicitly.
- All laboratory measurements were obtained in a single center using the same instrument and assay kit; therefore, outlier exclusion and batch‐effect correction were not performed. We have added the following statement to the Limitations section: “Because all data were generated under homogeneous conditions, no outlier removal or batch‐effect adjustment was applied, which may limit generalizability.”
- The timing of biomarker sample collection has been further specified in the Methods section.
- Patients who died before radiological evaluations could be performed or who were lost to follow‐up were excluded from the analysis and this added to data collection
- To demonstrate the absence of problematic collinearity, we have included a predictor correlation heat-map (Supplementary Fig. S2), which shows all pairwise correlations |r| < 0.7. We also computed variance inflation factors (VIFs) for the logistic regression model, and all VIF values were below 2, confirming that multicollinearity is not a concern.
- Figures, particularly the nomogram and calibration plots, remain difficult to interpret. Labels such as “points” and “risk score” are undefined, and the legends lack sufficient descriptive detail. Incorporating a summary visualization of variable contributions (e.g., a feature importance plot or heatmap) would improve clarity and accessibility.
We thank the Reviewer for the suggestion to include summary visualizations of variable contributions. In response, we have added two Supplementary Figures:
A correlation heatmap demonstrating the pairwise relationships among the five predictors and confirming the absence of problematic multicollinearity (Supplementary Fig. S1).
A range-adjusted feature importance barplot illustrating the relative impact of each predictor on primary resistance (Supplementary Fig. S2).,
All font inconsistencies in the nomogram have been corrected and standardized to improve legibility
We believe these additions enhance the transparency and interpretability of our model.
- Several statements in the Abstract, Results, and Conclusion imply the model’s clinical readiness. Given the exploratory nature of the analysis and the absence of prospective or external validation, these claims should be reworded more cautiously. The model may offer insights into hypothesis generation but should not be presented as a decision-support tool at this stage.
In this regard, the language has been tempered and the findings are now presented in a more hypothetical, hypothesis‐generating context.
- The revised Discussion includes additional references but still omits relevant recent studies on machine learning models and immune biomarker prediction in melanoma. Including these would better position the work within the current scientific landscape and acknowledge ongoing efforts in this domain.
At the reviewer’s suggestion, we have added a new paragraph in the Discussion outlining early approaches to primary resistance—covering genomic strategies, recent machine-learning models, and candidate biomarkers. To address the reviewer’s concerns about English language quality, we also employed MDPI’s Rapid English Editing service to correct and polish the manuscript.

Round 3
Reviewer 1 Report
Comments and Suggestions for Authors
The revised manuscript presents a predictive nomogram for primary resistance to PD-1 inhibitors in metastatic melanoma, utilizing retrospective clinical and hematologic parameters. The authors have made several meaningful improvements, including the addition of internal bootstrap validation, enhanced figure clarity, and some expansion of the discussion. These updates improve the structure and transparency of the work and are commendable. However, important limitations remain that must be addressed for the manuscript to meet the standards of a clinically applicable predictive tool.
The most significant unresolved issue is the absence of external validation. While the added internal validation strengthens the model slightly, it does not address concerns about generalizability or overfitting. The model’s utility for clinical decision-making remains speculative, and conclusions suggesting clinical applicability should be tempered throughout the manuscript. Additionally, the biological rationale behind the inclusion of specific predictors, such as globulin and BRAF status, is still insufficiently developed. A more mechanistic discussion linking these variables to immunotherapy resistance would greatly enhance the relevance and interpretation of the findings.
Key methodological details also require further clarification. The manuscript does not clearly describe how outliers and batch effects were handled, nor does it report whether logistic regression assumptions—such as linearity and absence of multicollinearity—were tested and met. The timing of biomarker collection relative to treatment initiation and the radiologic assessment procedures for defining resistance also remain vague and should be clearly described to ensure reproducibility.
Although figure presentation has improved, additional refinements would enhance interpretability. For instance, key terms such as “points” and “risk score” should be clearly defined within the figure legends. Language across the manuscript is generally comprehensible but would benefit from minor grammatical editing to improve precision and flow.
In conclusion, this manuscript provides an exploratory model that may be useful for hypothesis generation in the field of immunotherapy resistance. However, without external validation or prospective evaluation, the findings should not be framed as immediately actionable. Further refinement of the biological context, statistical rigor, and cautious interpretation would strengthen the manuscript considerably.
Comments on the Quality of English LanguageThe manuscript is generally understandable, but the English language and writing style require moderate revision to improve clarity and professionalism. Several sentences are awkwardly constructed, with issues in grammar, punctuation, and word choice.
Author Response
We sincerely thank the reviewer for their constructive feedback and appreciation of the improvements made to our manuscript. We fully acknowledge the remaining limitations and have addressed each of the reviewer’s comments point-by-point below.
1.The most significant unresolved issue is the absence of external validation. While the added internal validation strengthens the model slightly, it does not address concerns about generalizability or overfitting. The model’s utility for clinical decision-making remains speculative, and conclusions suggesting clinical applicability should be tempered throughout the manuscript. Additionally, the biological rationale behind the inclusion of specific predictors, such as globulin and BRAF status, is still insufficiently developed. A more mechanistic discussion linking these variables to immunotherapy resistance would greatly enhance the relevance and interpretation of the findings.
As correctly highlighted by the reviewer, external validation is crucial for evaluating the generalizability and clinical utility of predictive models. Unfortunately, due to the retrospective and single-center nature of our dataset, external validation was not feasible at this stage. We have explicitly acknowledged this critical limitation in the Discussion section and tempered our previous statements regarding the clinical applicability of the nomogram. It is now clearly stated that our model is exploratory and primarily aimed at hypothesis generation regarding resistance to immunotherapy.
Regarding the biological rationale, we agree that further explanation was needed. Although large-scale studies generally suggest that BRAF mutation status alone does not significantly reduce immunotherapy effectiveness, some studies have indicated that specific mutations, particularly those accompanied by resistance mutations like PTEN loss, may be associated with treatment resistance. Additionally, several studies indicate that specific immunoglobulin subtypes (such as high levels of IgG2) correlate positively with survival outcomes in melanoma patients undergoing immunotherapy. We have expanded our Discussion section to incorporate and discuss these findings.
2.Key methodological details also require further clarification. The manuscript does not clearly describe how outliers and batch effects were handled, nor does it report whether logistic regression assumptions—such as linearity and absence of multicollinearity—were tested and met. The timing of biomarker collection relative to treatment initiation and the radiologic assessment procedures for defining resistance also remain vague and should be clearly described to ensure reproducibility.
Regarding the handling of outliers, boxplot analyses were conducted, and no significant outliers affecting the outcomes were identified. This has now been clearly mentioned in the Methods section. All laboratory tests were performed at the same institution using identical laboratory equipment, thus batch effects were deemed negligible and this detail has been added to the methodology.
Concerning logistic regression assumptions, linearity was evaluated both visually and through the Box-Tidwell test, confirming that the linearity assumption was adequately met. Multicollinearity among predictors was assessed using variance inflation factor (VIF) tests, and all values were found to be below 2, indicating the absence of significant multicollinearity. Visual results and supporting data have been added as supplementary (non-published) materials. Additionally, the timing of biomarker collection (within the last month before treatment initiation) and the intervals for radiological assessments have now been explicitly stated in the data collection section.
3.Although figure presentation has improved, additional refinements would enhance interpretability. For instance, key terms such as “points” and “risk score” should be clearly defined within the figure legends. Language across the manuscript is generally comprehensible but would benefit from minor grammatical editing to improve precision and flow.
We have elaborated on the nomogram’s description to enhance its clarity and facilitate reader comprehension.
